# The role of the C$_2$A domain of synaptotagmin 1 in asynchronous neurotransmitter release

**Mallory C. Shields[1,2], Matthew R. Bowers[1,2], Hannah L. Kramer[1], McKenzie M. Fulcer[1], Lara C. Perinet[1], Marissa J. Metz[1,2], Noreen E. Reist ◉[1,2]***

**1** Department of Biomedical Sciences, Colorado State University, Fort Collins, Colorado, United States of America, **2** Molecular, Cellular, and Integrative Neuroscience Program, Colorado State University, Fort Collins, Colorado, United States of America

\* reist@colostate.edu

**Data Availability Statement:** All data associated with this manuscript are freely available at Mountain Scholar (http://dx.doi.org/10.25675/10217/204914).

## Abstract

Following nerve stimulation, there are two distinct phases of Ca$^{2+}$-dependent neurotransmitter release: a fast, synchronous release phase, and a prolonged, asynchronous release phase. Each of these phases is tightly regulated and mediated by distinct mechanisms. Synaptotagmin 1 is the major Ca$^{2+}$ sensor that triggers fast, synchronous neurotransmitter release upon Ca$^{2+}$ binding by its C$_2$A and C$_2$B domains. It has also been implicated in the inhibition of asynchronous neurotransmitter release, as blocking Ca$^{2+}$ binding by the C$_2$A domain of synaptotagmin 1 results in increased asynchronous release. However, the mutation used to block Ca$^{2+}$ binding in the previous experiments (aspartate to asparagine mutations, *syt$^{D-N}$*) had the unintended side effect of mimicking Ca$^{2+}$ binding, raising the possibility that the increase in asynchronous release was directly caused by ostensibly constitutive Ca$^{2+}$ binding. Thus, rather than modulating an asynchronous sensor, *syt$^{D-N}$* may be mimicking one. To directly test the C$_2$A inhibition hypothesis, we utilized an alternate C$_2$A mutation that we designed to block Ca$^{2+}$ binding without mimicking it (an aspartate to glutamate mutation, *syt$^{D-E}$*). Analysis of both the original *syt$^{D-N}$* mutation and our alternate *syt$^{D-E}$* mutation at the *Drosophila* neuromuscular junction showed differential effects on asynchronous release, as well as on synchronous release and the frequency of spontaneous release. Importantly, we found that asynchronous release is not increased in the *syt$^{D-E}$* mutant. Thus, our work provides new mechanistic insight into synaptotagmin 1 function during Ca$^{2+}$-evoked synaptic transmission and demonstrates that Ca$^{2+}$ binding by the C$_2$A domain of synaptotagmin 1 does not inhibit asynchronous neurotransmitter release *in vivo*.

## Significance statement

This study provides mechanistic insights into synaptotagmin function during asynchronous neurotransmitter release and supports a dramatically different hypothesis regarding the mechanisms triggering asynchronous vesicle fusion. Using two distinct C$_2$A mutations that block Ca$^{2+}$ binding, we report opposing effects on synchronous, spontaneous, and asynchronous neurotransmitter release. Importantly, our data demonstrate that Ca$^{2+}$ binding by the C$_2$A

**Funding:** NER is funded by the National Science Foundation (1257363, www.nsf.gov) and a Colorado State University (CSU) College of Veterinary Medicine and Biomedical Sciences award (CRC, https://vetmedbiosci.colostate.edu/). MCS is funded through the Vice President for Research Fellowship at CSU (https://www.research.colostate.edu/). Funders did not play a role in any aspect of the study.

**Competing interests:** The authors have declared that no competing interests exist.

domain of synaptotagmin does not regulate asynchronous release and thus disprove the current inhibition hypothesis. We propose a spatial competition hypothesis to explain these seemingly discordant results of the differing C$_2$A Ca$^{2+}$ binding mutations.

## Introduction

Fast, synchronous release is the large burst of neurotransmitter release that peaks within milliseconds (ms) of the arrival of the action potential. At most healthy synapses, the majority of release occurs during the synchronous phase [1]. Synaptotagmin 1, which contains two Ca$^{2+}$-binding domains, C$_2$A and C$_2$B [2], is essential for coupling Ca$^{2+}$ binding to efficient, synchronous release [3–6].

Asynchronous release can last from 10's of ms to 10's of seconds [7], and has been functionally implicated in synaptic plasticity [8–11] and development [12, 13]. While most synapses exhibit little to no asynchronous release, it is observed in several synaptic types [7]. Indeed, at some specialized synapses, such as certain hippocampal interneurons, asynchronous release is predominant [14].

In addition to being the Ca$^{2+}$ sensor for synchronous release, synaptotagmin 1 is proposed to regulate asynchronous release. Increases in asynchronous release are reported in synaptotagmin 1 null mutants [15, 16] and in a synaptotagmin 1 point mutant in which Ca$^{2+}$ binding by the C$_2$A domain is blocked [17]. Importantly, synchronous release remains intact in this C$_2$A point mutant, suggesting that Ca$^{2+}$ binding by C$_2$A is not needed for efficient synchronous release, but does play a role in preventing asynchronous neurotransmission. Together, these studies resulted in the inhibition hypothesis: that Ca$^{2+}$ binding by the C$_2$A domain of synaptotagmin is directly inhibiting asynchronous neurotransmitter release.

More recently, our group demonstrated that Ca$^{2+}$ binding by the C$_2$A domain *is required* for efficient synchronous release [4], contrary to previous studies [18–20]. The previous point mutations used to block Ca$^{2+}$ binding by C$_2$A all removed negative charge from the Ca$^{2+}$ binding pocket; key, negatively-charged aspartate residues (D) essential for coordinating Ca$^{2+}$ were replaced with neutral asparagines (N), *syt$^{D-N}$*. Since synaptotagmin 1 functions as an electrostatic switch [21, 22], removing negative charge may mimic Ca$^{2+}$ binding [18] and permit downstream effector interactions.

To directly test this hypothesis, we utilized our Ca$^{2+}$-binding mutation where an essential C$_2$A aspartate was mutated to a negatively-charged glutamate (E), *syt$^{D-E}$* [4]. This *syt$^{D-E}$* mutation maintains the negative charge of the pocket but prevents Ca$^{2+}$ binding by steric hindrance, resulting in an ~80% decrease in synchronous release. This finding demonstrated that an intact C$_2$B Ca$^{2+}$-binding domain is *only* sufficient to trigger the electrostatic switch in the absence of C$_2$A Ca$^{2+}$ binding *if* the negative charge of the C$_2$A Ca$^{2+}$-binding pocket is neutralized. Thus, the failure of *syt$^{D-N}$* mutations to impair synchronous release is a direct consequence of removing the electrostatic repulsion of the presynaptic membrane [4].

The current inhibition hypothesis, that a *syt$^{D-N}$* mutation fails to inhibit asynchronous release because it cannot bind Ca$^{2+}$, may be subject to the same misinterpretation. By comparing the *syt$^{D-N}$* mutation known to increase asynchronous release with our *syt$^{D-E}$* mutation, we directly test whether Ca$^{2+}$ binding by the C$_2$A domain of synaptotagmin 1 is required to regulate asynchronous release events *in vivo*. If C$_2$A Ca$^{2+}$ binding inhibits asynchronous release, the increased asynchronous release seen in the *syt$^{D-N}$* mutation should also occur in the *syt$^{D-E}$* mutation. However, if increased asynchronous release is a consequence of ostensibly constitutive Ca$^{2+}$ binding in *syt$^{D-N}$*, then the *syt$^{D-E}$* mutation should not result in increased asynchronous release. We now show that the *syt$^{D-N}$* and *syt$^{D-E}$* mutations had differential effects on each form of neurotransmitter release: spontaneous, synchronous, and asynchronous.

Importantly, the $syt^{D-E}$ mutation had no impact on asynchronous release, demonstrating that C$_2$A Ca$^{2+}$ binding does not regulate asynchronous neurotransmitter release.

## Materials and methods

### Drosophila strains

The aspartate to asparagine line used was $P[UAS\text{-}syt\ 1^{C2A\text{-}D3,4N}]$ (generously provided by Motojiro Yoshihara, University of Massachusetts Medical School, Worcester, MA, [17]) that encodes the synaptotagmin 1$^{D282N,D284N}$ mutant protein. The aspartate to glutamate line used was $P[UAS\text{-}syt^{D2E}]$ [4] that encodes the synaptotagmin 1D229E mutant protein. Expression of wild-type synaptotagmin from a transgene in an otherwise $syt1^{null}$ background does not provide full rescue of synchronous neurotransmitter release [23]. Therefore, a P[UAS-$syt1^{WT}$] transgenic line was used as the positive control in all experiments. GeneWiz (South Plainfield, New Jersey) synthesized cDNA of the *Drosophila* wild type *syt 1* gene, including some 5' and 3' untranslated sequence [24] to match the mutant *syt* transgenes as closely as possible, flanked by 5' EcoRI and 3' BglII restriction sites. This wild type transgene was placed under the control of the UAS promoter by directional subcloning into the pUAST-attB vector using the EcoRI and BglII restriction sites. The transgene was inserted in the attP2 landing site on the third chromosome in *Drosophila* using the PhiC31 targeted insertion system [25] by Genetivision (Houston, TX). The UAS/Gal4 system with an elav promoter was used to drive pan neuronal expression of all *syt 1* transgenes [26, 27]. The elavGal4 line used, $syt^{AD4}\ P[elavGal4]$, was generated by genetic recombination [28, 29]. To assess the functional significance of the mutations, all transgenes were expressed in the absence of endogenous synaptotagmin 1 by crossing them into the *syt 1null* mutant background, $syt^{AD4}$ [28]. As no gender selection was employed, experimental larvae included both males and females. The genotypes of experimental larvae used in all experiments were the following: *yw; $syt^{AD4}\ P[elavGal4]/syt^{AD4}$; $P[UASsyt\ 1^{WT}]/+$* (referred to as $P[syt^{WT}]$ or control), *yw; $syt^{AD4}\ P[elavGal4]/syt^{AD4}$; $P[UASsyt\ 1^{C2A\text{-}D3,4N}]/+$* (referred to as $P[syt^{D-N}]$), and *yw; $syt^{AD4}\ P[elavGal4]/syt^{AD4}$; $P[UASsyt\ 1^{D2E}]/+$* (referred to as $P[syt^{D-E}]$). All experimental lines are available upon request.

### Immunoblotting

The level of synaptotagmin expression was determined by western blot analysis using actin levels as a loading control. Third instar central nervous systems (CNSs) were dissected in HL3.1 saline (70 mM NaCl, 5 mM KCl, 1.0 mM CaCl$_2$, 4 mM MgCl$_2$, 10 mM NaHCO$_3$, 5 mM Trehalose, 115 mM sucrose, 5 mM HEPES, pH 7.2 [30]) where the CaCl$_2$ was omitted to decrease vesicle fusion events during the dissection. Individual CNSs were sonified with five 0.3 s pulses at 1 Hz using a Branson Sonifier 450 (VWR Scientific, Winchester, PA) in Laemmli buffer (Bio-Rad, Hercules, CA) containing 5% β-mercaptoethanol. Each sample was separated by SDS-PAGE with 15% acrylamide, transferred to Immobilon membranes (Millipore, Bedford, MA), and washed in blocking solution [5% milk, 4% normal goat serum (NGS, Fitzgerald Industries International, Acton, MA), 1% bovine serum albumin (BSA, Millipore-Sigma, Burlington, MA), and 0.02% NaN$_3$ in PBS-Tween (phosphate-buffered saline, 137 mM NaCl, 1.5 mM KH$_2$PO$_4$, 2.7 mM KCl, 8.1 mM Na$_2$HPO$_4$ containing 0.05% Tween 20, Fisher BioReagents, Fair Lawn, NJ)]. Membranes were probed overnight at 4˚C with a 1:2,500 dilution of anti-synaptotagmin antibody, Dsyt-CL1 [3] and 1:10,000 dilution of anti-actin antibody, (MAB 1501, Millipore Bioscience Research Reagents, Billerica, MA) in PBS-Tween containing 10% NGS and 0.02% NaN$_3$. Membranes were washed in PBS-Tween for 1–3 hours, then probed with a 1:5,000 dilution of Peroxidase-conjugated AffiniPure Goat Anti-Rabbit IgG (Jackson ImmunoResearch, West Grove, PA) and a 1:5,000 dilution of Peroxidase-conjugated

AffiniPure Donkey Anti-Mouse IgG (Jackson ImmunoResearch, West Grove, PA) in PBS-Tween containing 10% NGS for 1 hour at room temperature. An Epichemi[3] Darkroom with Labworks Imaging Software (UVP BioImaging, Upland, CA) was used to visualize the protein bands. Quantification: for each blot, synaptotagmin/actin ratios were calculated and normalized to the mean synaptotagmin/actin ratio of the control lanes to permit comparison of synaptotagmin signals between blots. Outliers in actin levels were excluded from analysis. Statistical significance was determined using student t-tests.

## Immunolabeling

To visualize the localization of transgenic synaptotagmin, third instar larvae were dissected in $Ca^{2+}$-free HL3.1 saline and fixed in PBS containing 4% formaldehyde for 1 hour. Samples were probed overnight at 4˚C in a 1:400 dilution of Dsyt-CL1 in dilution media [PBS with 0.1% Triton (PBST), 1% BSA, and 1% NGS (Jackson ImmunoResearch, West Grove, PA)]. Samples were washed in PBST for 1–3 hours, incubated in dilution media containing a 1:400 dilution of Alexa Fluor 488 goat anti-rabbit antibody (Invitrogen, Carlsbad, CA) for 1 hour at room temperature, washed in PBST for one hour, and mounted on microscope slides in Citifluor (Ted Pella, Redding, CA). Images of neuromuscular junctions of abdominal muscles 6/7 in segments 3 and 4 were acquired using a Zeiss 880 light scanning microscope, a 40X objective, and Zeiss Zen 2.1 acquisition software, version 11,0,3,190.

## Electrophysiological experiments and analyses

All electrophysiological events were collected with an Axoclamp 2B amplifier (Molecular Devices, Sunnyvale, CA), a Powerlab 4/35 A/D converter (ADInstuments, Sydney, Australia), and LabChart software (ADInstruments, Sydney, Australia). 10–20 MΩ intracellular electrodes were pulled using a Sutter model P-97 micropipette puller (Novato, CA) and filled with 3 parts 2 M $K_3C_6H_5O_7$ to 1 part 3 M KCl. Third instar larvae were dissected in $Ca^{2+}$-free HL3.1 saline to expose the body wall musculature and the central nervous systems were removed. All recordings of synchronous, asynchronous, and spontaneous neurotransmitter release were made from muscle 6 of abdominal segments 3 and 4 in HL3.1 containing 1.0 mM $Ca^{2+}$. The resting potential was held at -55 mV by applying no more than 1 nA of current.

## Evoked release

Ten excitatory junction potentials (EJPs) were collected at 0.04 Hz using a stimulating electrode filled with HL3.1 saline. The mean response was calculated for each fiber and the mean amplitude from 10–11 fibers per genotype is reported. Statistical significance was determined using a one-way ANOVA with Dunnett's correction.

Miniature excitatory junction potentials (mEJPs) were acquired for 3 minutes in HL3.1 saline. Events were identified manually after recordings had been randomized and blinded to the researcher for analysis. Mean mEJP amplitudes were determined from 50 consecutive events/fiber, beginning at minute two of each recording. Statistical significance was determined using the Kruskal-Wallis test. To determine mEJP frequency, all mEJP events between the 2nd and 3rd minute of recording from each fiber were counted. Statistical significance of mEJP frequency was determined using a one-way ANOVA with Dunnett's correction.

## Paired pulse

For paired pulse analysis, muscle fibers were stimulated twice with a 20 ms interpulse interval. Paired pulse ratios were determined by dividing the EJP amplitude of the second response by

that of the first. The first response was measured as the potential change from baseline to the first peak, and the second response amplitude was measured as the potential change from the trough between the responses and the amplitude of the second peak. Mean paired pulse ratios from 13–20 fibers/genotype were compared between genotypes using a one-way ANOVA with Dunnett's correction.

### Hypertonic sucrose stimulation

For hypertonic solution stimulations, a puff application of modified Ca$^{2+}$-free HL3.1 saline containing 0.3 M sucrose was administered to the junctional region of muscles 6/7 in abdominal segments 3 and 4 of third instars using a PicoSpritzer III (Parker Hannifan, Pine Brook, NJ). The puff application was administered for 5 s at ~ 5 pounds per square inch. Recordings were acquired in a bath solution of Ca$^{2+}$-free HL3.1 saline. All recordings were randomized and blinded to the researcher for analysis. mEJP events were counted manually for 70 s starting 10 s prior to sucrose application, to permit calculation of the baseline mEJP frequency. Events were parsed into 1 s bins. To determine the response to the hypertonic solution in each genotype, the mean mEJP frequency during the 40 s following the initiation of the sucrose application was calculated. Statistical significance was determined using a Kruskal-Wallis test with Dunn's correction.

### Two-electrode voltage clamp

Since the individual vesicle fusion events typical of asynchronous release cannot be resolved by current clamp recordings during the first 10's of milliseconds following a large synchronous response due to non-linear summation, we used two-electrode voltage clamp (TEVC) for these experiments. Synaptic currents were recorded according to standard protocols [31]. Any muscle fiber with an input resistance < 5 MΩ following insertion of a second intracellular electrode of 10–15 MΩ resistance was excluded. Only recordings that attained a ≥ 90% clamp of membrane voltage were included in the final analyses. Fibers were held at -55 mV using no more than 1 nA of current. 12–17 fibers/genotype were stimulated 5 times at 0.2 Hz in HL3.1 saline.

For asynchronous release event frequencies, all release events from 280 ms before stimulation to 580 ms after stimulation were counted manually in each of the 5 traces/fiber and parsed into 20 ms bins. A single, multi-quantal synchronous response occurred during the 0–20 ms bin following stimulation and was counted as 1 event. Paired t-tests were used to compare event frequencies within each genotype during the prestimulation time period (the 280 ms prior to stimulation) with that of the asynchronous release time period (20–300 ms after stimulation) and with that of the recovery time period (300–580 ms after stimulation). To compare asynchronous event frequencies across genotypes while controlling for differences in the frequency of spontaneous release, the number of events during the prestimulation time period was subtracted from the number of events during the asynchronous time period for each trace and the mean difference was calculated for each genotype. Statistical significance between genotypes was determined using a Kruskal-Wallis test with Dunn's correction.

Charge transfer was quantified as an independent means of measuring asynchronous release. The average charge transfer of five traces from each fiber was calculated to diminish noise. The average resting membrane potential was calculated from the mean potential during the 280 ms prior to stimulation and was used to determine the beginning of the response post stimulation. Total charge transfer represents the total evoked response and was measured from the beginning of the response to 300 ms post stimulation. To assess asynchronous release, we divided the total response into synchronous, beginning of response to 20 ms, and

asynchronous, 20–300 ms, components based on completion of the synchronous component in control traces [17]. Statistical significance between genotypes was determined using a one-way ANOVA with Dunnett's correction.

## Experimental design and statistical analysis

Statistical analyses were performed using Prism 7 (GraphPad software, La Jolla, CA). Each data set was collected from at least 9 samples for all experiments. Unpaired student t-tests with t values were used to compare datasets with Gaussian distributions between two genotypes. Paired t-tests were employed when comparing values within a given electrophysiological trace. One-way ANOVAs with F values and Dunnett's correction were used to compare datasets with Gaussian distributions across all 3 genotypes. Kruskal-Wallis tests with Dunn's correction were used to compare datasets with non-Gaussian distributions across all 3 genotypes. An alpha p-value of 0.05 was considered significant. Raw dataset is posted at https://lib.colostate.edu/find/csu-digital-repository/.

## Results

### Synaptotagmin mutations

To test the function of Ca$^{2+}$ binding by the C$_2$A domain of synaptotagmin during vesicle fusion events, we completed the first direct comparison of two disparate mutations that both block Ca$^{2+}$ binding (Fig 1A). Mutating the third and fourth of the Ca$^{2+}$-binding aspartates to asparagines (*syt$^{D-N}$*, Fig 1A) blocks Ca$^{2+}$ binding by removing key negative charges required to coordinate Ca$^{2+}$ [20, 32, 33]. Mutating the second aspartate to a glutamate (*syt$^{D-E}$*, Fig 1A) in C$_2$A blocks Ca$^{2+}$ binding by steric hindrance while maintaining the negative charge of the pocket [4]. In all experiments, synaptotagmin 1 was expressed as a transgene (*P[syt]*) in the absence of native synaptotagmin 1.

### Transgenic synaptotagmin expression and targeting

To compare levels of synaptotagmin expression between *P[syt$^{WT}$]*, *P[syt$^{D-N}$]*, and *P[syt$^{D-E}$]*, western analysis was performed on the central nervous systems (CNSs) of individual third instar larvae. Synaptotagmin levels were normalized to actin levels. As previously shown [4, 17], there were no significant differences in expression of the synaptotagmin transgenes. Mean expression ± standard error of the mean (SEM) in *P[syt$^{D-N}$]* was 104.6 ± 8.4% of control (Fig 1B, left, p = 0.70, t(8) = 1.24, student t-test). Mean expression in *P[syt$^{D-E}$]* was 112.6 ± 10.4% of control (Fig 1B, right, p = 0.29, t(9) = 2.98, student t-test). Staining of third instar larval body wall preparations with an anti-synaptotagmin antibody demonstrated that transgenic synaptotagmin is appropriately concentrated at the neuromuscular junction in all three genotypes (Fig 1C and [4, 17]).

### Opposite effects of Ca$^{2+}$-binding mutants on synchronous release amplitude and spontaneous release frequency

Both the *syt$^{D-N}$* and *syt$^{D-E}$* mutations block Ca$^{2+}$ binding by the C$_2$A domain (Fig 1A and [4, 20, 32–34]). We determined EJP amplitude in larvae under our recording conditions (Fig 2A and 2B) to directly compare these mutants and to ensure that our TEVC recordings attained a ≥ 90% clamp of this response. The *P[syt$^{D-N}$]* mutant exhibited efficient synchronous neurotransmitter release at larval neuromuscular junctions (Fig 2A and 2B), as previously reported by voltage clamp in embryonic preparations [17]. However, the *P[syt$^{D-E}$]* mutant did not (Fig 2A and 2B and [4]). ANOVA analysis showed a significant difference between the three

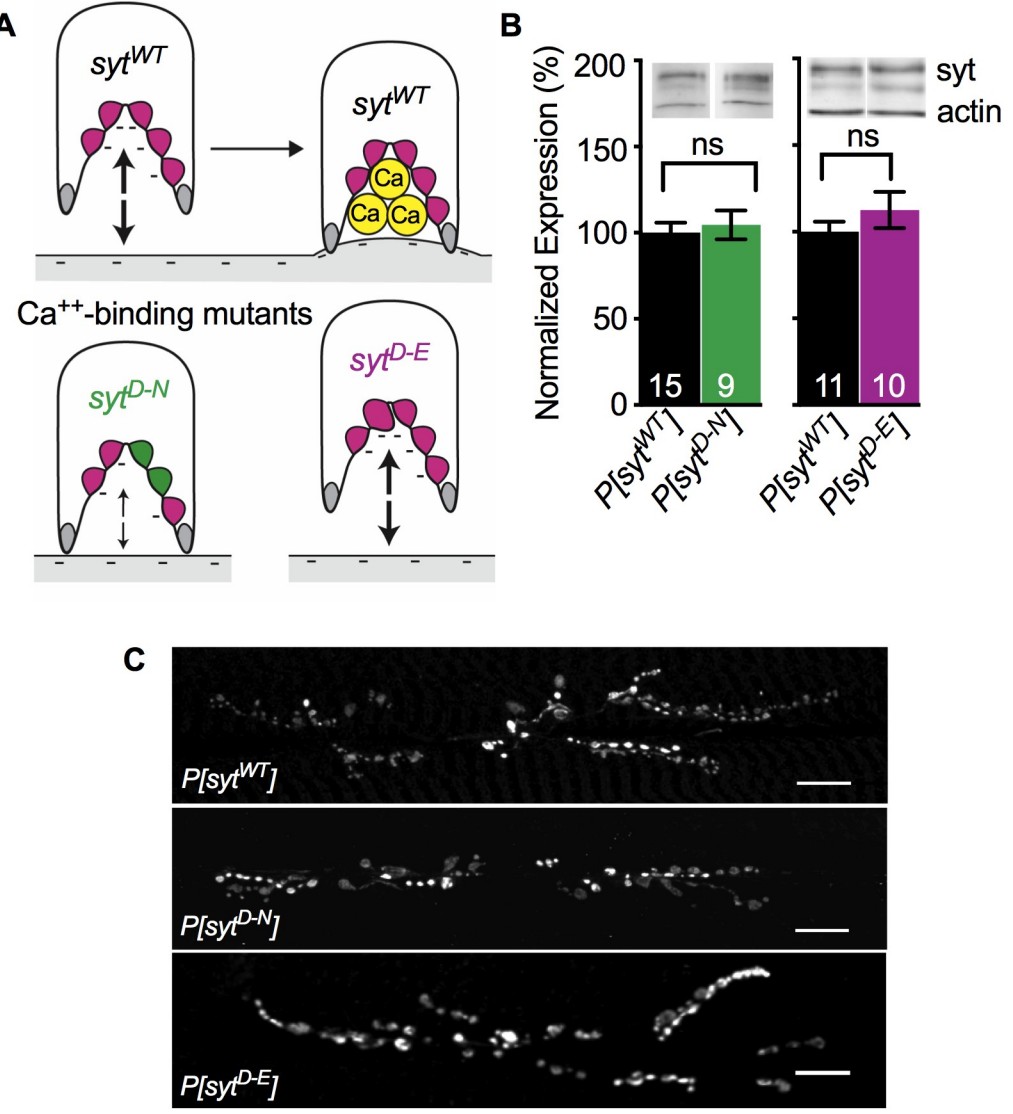

**Fig 1. Ca²⁺ binding mutants expressed and localized correctly. A**. Schematic depiction of the C₂A domain of wild type and mutant synaptotagmin and their postulated interactions with the negatively-charged presynaptic membrane. **A. Top**, The C₂A Ca²⁺-binding pocket of wild type synaptotagmin (*syt^WT^*). Prior to Ca²⁺ entry (left), 5 negatively charged (magenta) aspartate residues repel the negatively-charged presynaptic membrane (large arrows). Ca²⁺ binding neutralizes the negative charge of the pocket (right), resulting in the penetration of the presynaptic membrane by hydrophobic residues (grey). **A. Below**, By replacing two aspartate residues with neutral (green) asparagines, the *syt^D-N^* mutation in C₂A blocks Ca²⁺ binding and partially neutralizes the negative charge of the pocket. Importantly, this partial neutralization also decreases the electrostatic repulsion of the presynaptic membrane (small arrows), which may mimic Ca²⁺ binding. By replacing one aspartate residue deep in the Ca²⁺-binding pocket with a larger, negatively-charged glutamate residue (magenta, larger), the *syt^D-E^* mutation in C₂A blocks Ca²⁺ binding by steric hindrance while maintaining electrostatic repulsion of the presynaptic membrane (large arrows). **B. Above**, Representative western blots showing expression levels of synaptotagmin and actin from individual larval CNSs of each genotype. **B. Below**, quantification of *P[syt^WT^]* vs. *P[syt^D-N^]* (left) and *P[syt^WT^]* vs. *P[syt^D-E^]* (right). All measurements were normalized to actin levels. Both mutant lines exhibited levels of synaptotagmin expression similar to control. ns = not significant. **C**. Anti-synaptotagmin labeling of third instar body wall musculature demonstrated that transgenic synaptotagmin is appropriately concentrated at the neuromuscular junction in all genotypes. Scale bars = 20 μm.

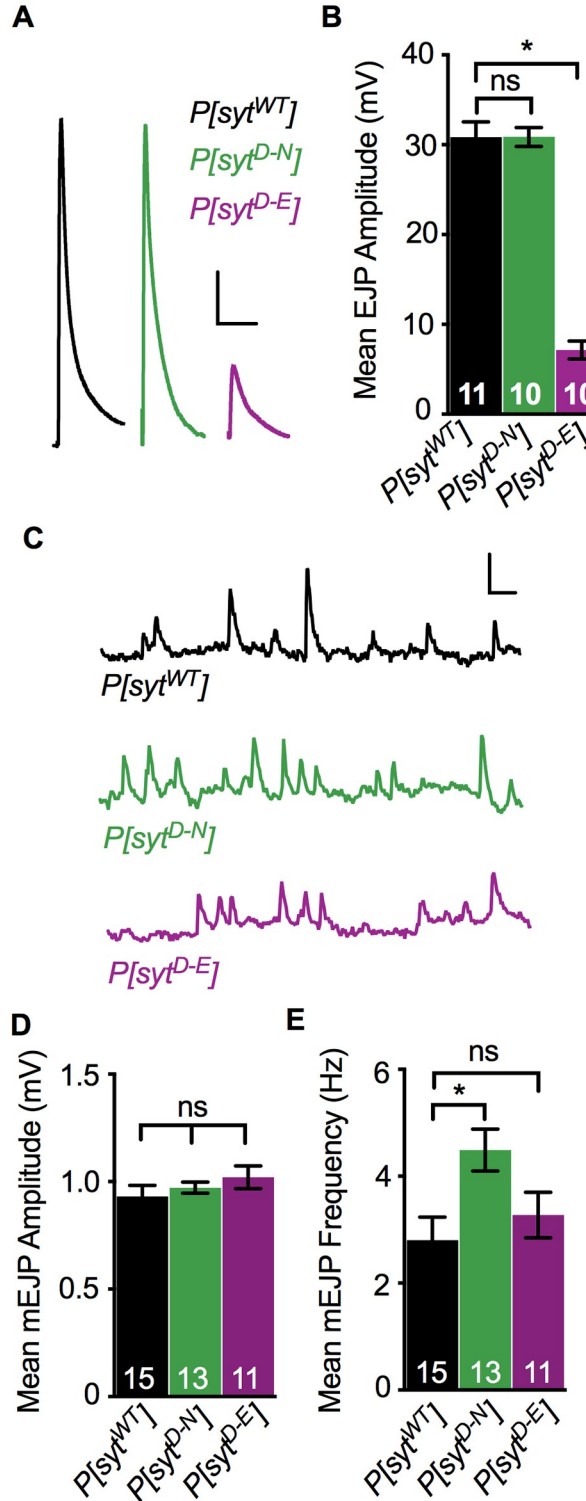

**Fig 2. Ca$^{2+}$ binding mutants had differential effects on evoked and spontaneous release. A**. Representative EJP traces from $P[syt^{WT}]$, $P[syt^{D-N}]$, and $P[syt^{D-E}]$. Scale bars = 5 mV, 0.1 s. **B**. Mean EJP amplitude in $P[syt^{D-N}]$ was unimpaired, but in $P[syt^{D-E}]$, it was significantly decreased compared to control (*p < 0.0001). **C**. Representative mEJP traces from $P[syt^{WT}]$, $P[syt^{D-N}]$, and $P[syt^{D-E}]$ showing 3 consecutive seconds of spontaneous mEJPs. Scale bars = 1 mV, 0.2 s. **D**. Mean mEJP amplitude was similar among genotypes. **E**. Mean mEJP frequency was increased in $P[syt^{D-N}]$ (*p = 0.01) but unchanged in $P[syt^{D-E}]$ relative to $P[syt^{WT}]$. Recorded in HL3.1 containing 1.0 mM Ca$^{2+}$, all error bars depict SEM, and n's within bars represent number of muscle fibers tested. ns = not significant.

genotypes ($p < 0.0001$, $F(2, 28) = 105.2$). There was no significant difference between the mean EJP amplitude in *P[syt^WT]* (30.85 ± 1.69 mV, mean ± SEM) and *P[syt^D-N]* (Fig 2B, 30.88 ± 1.05 mV, p = 0.99, Dunnett's correction). The mean EJP amplitude in *P[syt^D-E]* was significantly reduced (Fig 2B, 7.15 ± 1.01 mV, $p < 0.0001$, Dunnett's correction). Thus, as seen previously [4], the *P[syt^D-E]* mutation resulted in ~80% decrease in neurotransmitter release. These results confirm that $Ca^{2+}$ binding by the $C_2A$ domain is critical for efficient synchronous neurotransmitter release and support the hypothesis that the *syt^D-N* mutation participates in triggering membrane fusion by mimicking $Ca^{2+}$ binding (Fig 1A).

We also compared the effect of these mutations on spontaneous neurotransmitter release, mEJPs (Fig 2C–2E). We found no significant differences in mEJP amplitudes in any genotype (Fig 2D, p = 0.19, Kruskal-Wallis Test). Mean mEJP amplitude ± SEM in *P[syt^WT]* was 0.93 ± 0.05 mV, in *P[syt^D-N]* was 0.97 ± 0.03 mV, and in *P[syt^D-E]* was 1.02 ± 0.05 mV. Thus, the decrease in synchronous evoked release in the *P[syt^D-E]* mutant was not due to a change in quantal size [4].

The effect of these mutations on the frequency of mEJP events is consistent with synaptotagmin's role as a clamp to prevent spontaneous fusion events. In synaptotagmin null mutants, the frequency of mEJPs is significantly increased compared to wild type [6, 35–37]. Similar to results from cultured mammalian neurons [38], the *P[syt^D-N]* mutant also exhibited an increased rate of spontaneous fusion events at *Drosophila* embryonic neuromuscular junctions [17]. We verified this effect at larval *Drosophila* neuromuscular junctions where we found statistically significant differences in mEJP frequency among the three genotypes (Fig 2E, p = 0.02, $F(2, 36) = 4.45$, ANOVA). The mEJP frequency in the *P[syt^D-N]* mutant was 4.49 ± 0.39 Hz (mean ± SEM), which was significantly increased compared to *P[syt^WT]* (2.81 ± 0.43 Hz, p = 0.01, Dunnett's correction). In contrast, mEJP frequency in *P[syt^D-E]* was similar to control (3.28 ± 0.43 Hz, p = 0.66, Dunnett's correction, and see [4]). Thus, the negative charge of the $Ca^{2+}$-binding pocket is the key characteristic of synaptotagmin required to clamp spontaneous fusion events. The differential effect of these mutations on both spontaneous and synchronous events supports synaptotagmin's role as an electrostatic switch, where electrostatic repulsion between its C2 domains and the presynaptic membrane is required to prevent vesicle fusion (see Fig 1A).

## Vesicles remain fusion competent

A decrease in the number of fusion competent synaptic vesicles could explain the decrease in neurotransmitter release in the *P[syt^D-E]* mutant. Hypertonic solutions cause vesicles to fuse with the presynaptic membrane in a $Ca^{2+}$-independent manner [39, 40]. To estimate the number of fusion competent vesicles in each transgenic line, we puff applied a 0.3 M sucrose solution to the neuromuscular junction (Fig 3A–3C). To control for the increase in spontaneous mEJP frequency in *P[syt^D-N]* (see Fig 2D), event frequencies were normalized to the mean mEJP frequency prior to sucrose application (Fig 3C). All three genotypes displayed similar increases in neurotransmitter release events during the sucrose response (Fig 3C, mean fold increase ± SEM between 0–40 s after sucrose application onset for *P[syt^WT]* = 2.59 ± 0.15, for *P[syt^D-N]* = 2.54 ± 0.08, and for *P[syt^D-E]* = 2.94 ± 0.13, p = 0.07, Kruskal-Wallis test; comparing *P[syt^WT]* to *P[syt^D-N]*, p = 0.99, and comparing *P[syt^WT]* to *P[syt^D-E]*, p = 0.13, Dunn's correction). Therefore, the decrease in synchronous release in the *P[syt^D-E]* mutant is not the result of a decrease in the number of fusion competent vesicles.

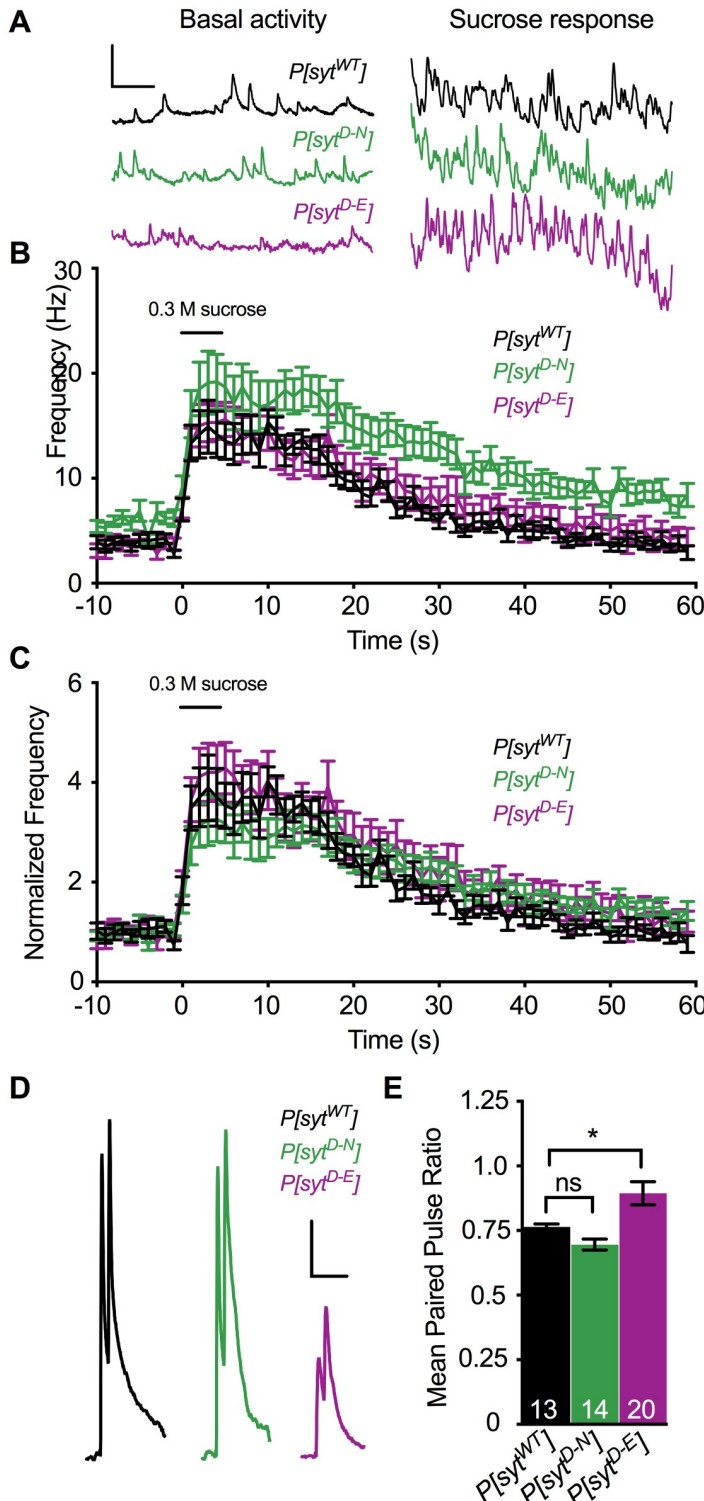

**Fig 3. Ca$^{2+}$ binding mutants had no impact on the number of fusion competent vesicles, but had differential effects on release probably. A**. Representative traces of event frequency before, during, and after sucrose-stimulated neurotransmitter release from $P[syt^{WT}]$, $P[syt^{D-N}]$, and $P[syt^{D-E}]$. Scale bars = 2 mV, 0.5 s. **B**. Mean event frequencies over time in response to a 5 s application of a hypertonic sucrose solution (n = 11 fibers for each genotype). **C**. Mean event frequencies over time normalized to the basal mEJP frequency prior to sucrose application. No statistically significant changes were found among genotypes during the sucrose response. The black bar above the traces in B,C

represents the 5 s sucrose application. **D**. Representative paired pulse traces with a 20 ms interpulse interval from *P [syt^WT]*, *P[syt^{D-N}]*, and *P[syt^{D-E}]*. Scale bars = 5 mV, 0.1 s. **E**. There was a significant increase in the paired pulse ratio in *P[syt^{D-E}]* compared to control (*p < 0.0001). There was no significant change in paired pulse ratios between control and *P[syt^{D-N}]*. Recorded in HL3.1 containing 1.0 mM Ca$^{2+}$, error bars are SEM, and n's within bars represent number of fibers tested. ns = not significant.

## Opposite effects of Ca$^{2+}$-binding mutants on release probability

Another potential explanation for the decrease in synchronous neurotransmitter release in the *P[syt^{D-E}]* mutant is a decrease in presynaptic release probability. Therefore, we compared paired pulse ratios in each genotype since an increase in the paired pulse ratio is correlated to a decrease in release probability [41]. Muscle fibers were stimulated with two pulses at an interpulse interval of 20 ms (Fig 3D and 3E, p = 0.0006, F(2, 44) = 8.76, ANOVA). The mean response (mV) ± SEM to the first pulse in *P[syt^WT]* = 29.72 ± 0.78, in *P[syt^{D-N}]* = 28.48 ± 1.24, and in *P[syt^{D-E}]* = 10.18 ± 0.71. The response to the second stimulation was 22.66 ± 0.57 mV in *P[syt^WT]*, 19.71 ± 0.89 mV in *P[syt^{D-N}]*, and 8.77 ± 0.5 mV in *P[syt^{D-E}]*. While the paired pulse ratio in *P[syt^{D-N}]* (0.70 ± 0.02, mean ± SEM) was similar to that in *P[syt^WT]* (0.76 ± 0.01, p = 0.35, Dunnett's correction), the paired pulse ratio in *P[syt^{D-E}]* (0.89 ± 0.05, p = 0.02, Dunnett's correction) was significantly larger. This increase in paired pulse ratio indicates that the *syt^{D-E}* mutation resulted in a decrease in release probability.

## Opposite effects of Ca$^{2+}$-binding mutants on asynchronous release

To quantitatively assess the timing of neurotransmitter release, we counted individual release events that occurred between 280 ms prior to stimulation and 580 ms after stimulation (Fig 4, Table 1, and see [16, 17]). Latency histograms of the mean number of events/stimulation before and after the stimulus are shown (Fig 4A–4C, right). A single, multi-quantal, synchronous response occurred during the first 20 ms following stimulation (Fig 4A–4C). The time course of Ca$^{2+}$-evoked asynchronous release was defined as 20–300 ms post stimulation (Fig 4A, right Asynch, Table 1 Mean Async), as event frequency was not elevated in any genotype during the 300–580 ms period post stimulation (Fig 4A, right Recovery, Table 1 Mean Recovery) compared to the values during the 280 ms prior to stimulation [(Fig 4A, right Prestim, Table 1 Mean Prestim) *P[syt^WT]*, p = 0.10, t(84) = 1.68; *P[syt^{D-N}]*, p = 0.13, t(59) = 1.54; *P [syt^{D-E}]*, p = 0.26, t(64) = 1.14; paired t-tests]. While control and the *P[syt^{D-E}]* mutant both showed a trend toward an increase in release events during the 20–300 ms asynchronous time window compared to prestim time window, these trends were not statistically significant (Fig 4D, Table 1, 25% increase for control, p = 0.11, t(84) = 1.62 and 11% increase for *P[syt^{D-E}]*, p = 0.52, t(64) = 0.65, paired t-tests). In contrast, the *P[syt^{D-N}]* mutant showed a robust and significant 67% increase in asynchronous release (Fig 4D, Table 1, p = 0.001, t(59) = 3.46, paired t-test).

Consistent with the increased mEJP frequency reported in voltage traces (Fig 2E), *P[syt^{D-N}]* exhibited an increased mEJC frequency prior to stimulation compared to *P[syt^WT]* (**, Fig 4D, Table 1, p = 0.03, F(2, 207) = 2.97, ANOVA; p = 0.01, Dunnett's correction), while *P[syt^{D-E}]* did not (p = 0.39, Dunnett's correction). To directly compare Ca$^{2+}$-stimulated, asynchronous release while controlling for the variable rate of mEJC frequency among genotypes, Prestim events were subtracted from Async events for each individual trace. Comparison of the mean difference [Fig 4E, Table 1 last column, Mean (Async—Prestim)/Trace] revealed significant differences in asynchronous release between genotypes (Fig 4E, p = 0.03, Kruskal-Wallis test). Consistent with previous reports [17], the *P[syt^{D-N}]* mutant exhibited an increase in the

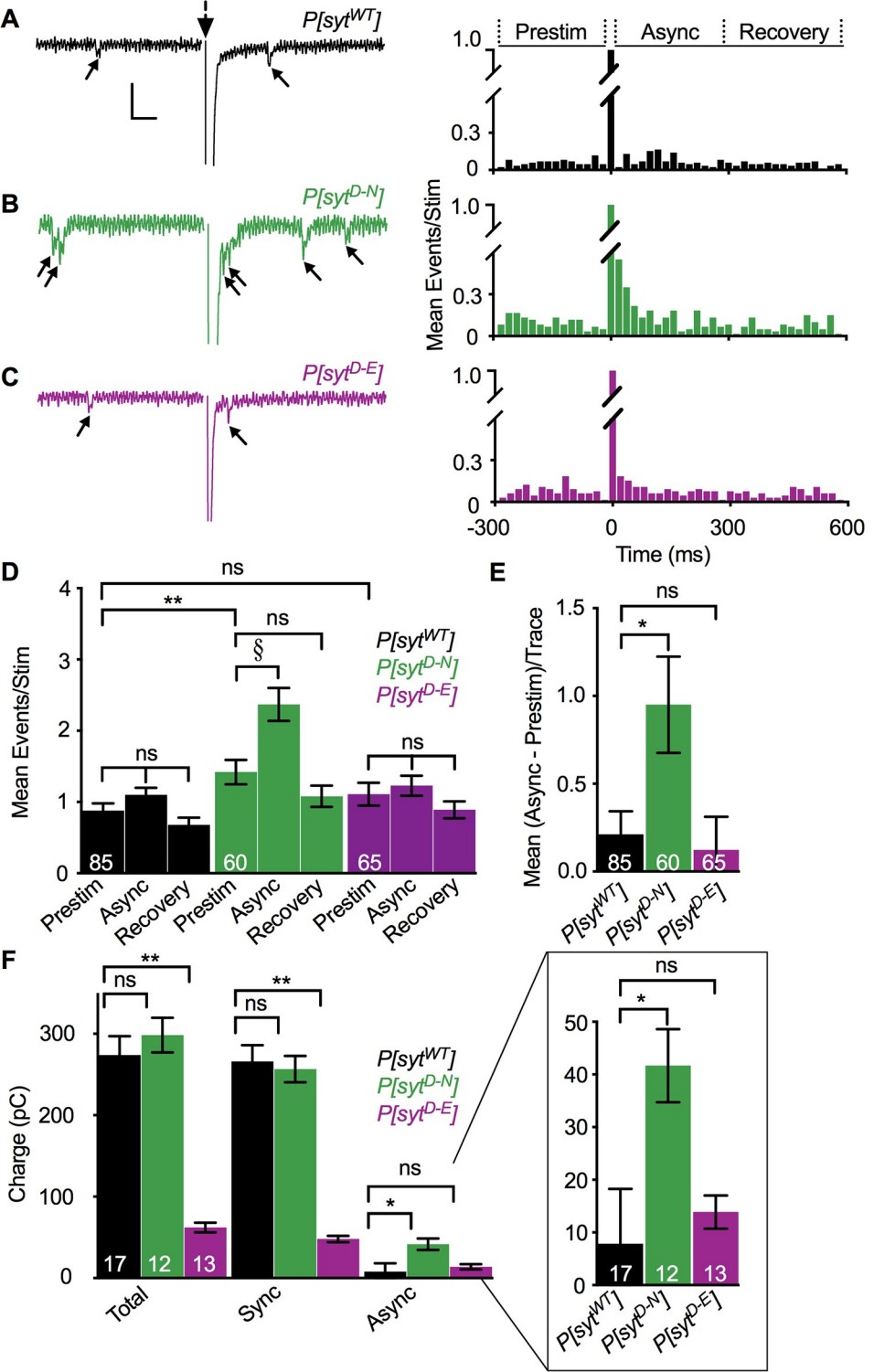

**Fig 4. Asynchronous release was increased in the *P[syt^D-N]* mutant but not the *P[syt^D-E]* mutant. A-C Left**.
Representative traces from *P[syt^WT]*, *P[syt^D-N]*, and *P[syt^D-E]* recorded in HL3.1 containing 1.0 mM Ca²⁺ showing
events between 280 ms before to 300 ms after stimulation (large dotted arrow). Stimulation artifact removed for clarity.
Individual release events before and after the large, multi-quantal synchronous response are indicated (small arrows).
Scale bars = 1 nA, 0.04 s. **Right**. Latency histograms. Data were parsed into 20 ms bins from 280 ms before to 580 ms
after single stimulations and the mean number of events/stimulation was plotted in each bin for all genotypes. **D**. Mean

events/stimulation during the 280 ms before stimulation (Prestim), the 20–300 ms after stimulation, asynchronous release period (Async), and the 300–580 ms after stimulation (Recovery) were graphed for each genotype. *P[syt^{D-N}]* larvae exhibited a significant increase in asynchronous release compared to Prestim (§p = 0.001). Control and the *P [syt^{D-E}]* mutant did not. No significant differences were found in any genotype when comparing Prestim and Recovery time periods. *P[syt^{D-N}]* also exhibited a significant increase in mEJC frequency (Prestim) compared to control (**p = 0.01). The *P[syt^{D-E}]* mutant did not. **E.** Compared to control, the *P[syt^{D-N}]* mutant exhibited a significant increase in asynchronous release corrected for mEJC frequency [Mean (Async—Prestim)/Trace, *p = 0.04], but the *P [syt^{D-E}]* mutant did not. **F.** Compared to control, the *P[syt^{D-N}]* mutant displayed no change in charge transfer during the total stimulated response (Total, green vs black) or during the synchronous phase of release (Sync, green vs black), but did display significantly greater charge transfer during the asynchronous phase of release (Async, green vs black, *p = 0.01). Conversely, the *P[syt^{D-E}]* mutant displayed no change in asynchronous release (Async, magenta vs black), but did display significantly less charge transfer during the total stimulated response and during the synchronous phase of release (Total and Sync respectively, magenta vs black **p < 0.0001). **F Inset**. Async data magnified for clarity. The n's within bars represent number of fibers tested. All error bars represent SEM. ns = not significant.

number of asynchronous release events compared to control (*, Fig 4E, Table 1, p = 0.04, Dunn's correction). Importantly, the *P[syt^{D-E}]* mutant exhibited no increase in asynchronous release events compared to control (Fig 4E, Table 1, p > 0.99, Dunn's correction).

Charge integrals were calculated as an independent method to quantify both synchronous and asynchronous release. Differences in total charge transfer were determined (Fig 4F, Total, p < 0.001, F(2, 37) = 6.404, ANOVA). Total charge transfer in the *P[syt^{D-N}]* mutant (298.4 ± 21.4 pC, mean ± SEM) was similar to that in control (273.9 ± 23.4 pC, p = 0.60, Dunnett's correction). However, total charge transfer was significantly decreased in the *P[syt^{D-E}]* mutant (62.2 ± 6.0 pC, p < 0.0001, Dunnett's correction) compared to control. Similar to the EJP amplitude analysis (Fig 2B), significant differences in charge transfer during the synchronous phase of release were found across genotypes (Fig 4F, Sync, p < 0.001, F(2, 39) = 7.841, ANOVA). The synchronous response in the control (266.1 ± 20.0 pC, mean ± SEM) was similar to that in the *P[syt^{D-N}]* mutant (256.7 ± 16.2 pC, p = 0.89, Dunnett's comparison) yet it was significantly decreased in the *P[syt^{D-E}]* mutant (48.0 ± 3.8 pC, p < 0.001, Dunnett's comparison). Similar to the event frequency analysis (Fig 4E), analysis of the charge transfer during the asynchronous phase also exhibited significant differences (Fig 4F, Asynch, inset at expanded resolution, p = 0.02, F(2, 39) = 1.345, ANOVA). Just as seen by the event frequency analysis (Fig 4E), the asynchronous phase of release was increased in the *P[syt^{D-N}]* mutant (41.7 ± 6.9 pC, mean ± SEM) relative to control (7.8 ± 10.4 pC, p = 0.01, Dunnett's correction). Importantly, there was no difference in the asynchronous phase of release in the *P[syt^{D-E}]* mutant

**Table 1. Mean number of events/stimulation across genotypes.**

| | Events/Stimulation | | | |
|---|---|---|---|---|
| | **Mean Prestim -280 to 0 ms** | **Mean Asynch 20 to 300 ms** | **Mean Recovery 300 to 580 ms** | **Mean $\left(\frac{Asynch-Prestim}{Trace}\right)$** |
| *P[syt^{WT}]* | 0.88 ± 0.10 | 1.09 ± 0.10 | 0.68 ± 0.10 | 0.21 ± 0.13 |
| *P[syt^{D-N}]* | 1.42 ± 0.17** | 2.37 ± 0.23§ | 1.08 ± 0.15 | 0.95 ± 0.27* |
| *P[syt^{D-E}]* | 1.11 ± 0.16 | 1.23 ± 0.14 | 0.89 ± 0.12 | 0.12 ± 0.19 |

Values are shown for the 280 ms pre stimulation (Mean Prestim), the 20–300 ms asynchronous release period (Mean Async), the recovery time period (Mean Recovery), and the mean difference between the Async and Prestim time periods calculated for each trace [Mean (Async—Prestim)/Trace] for *P[syt^{WT}]*, *P[syt^{D-N}]*, and *P[syt^{D-E}]*. *P [syt^{D-N}]* was the only genotype to display a significant increase in asynchronous release (Mean Async events compared to Mean Prestim events, §p = 0.001). *P[syt^{D-N}]* also exhibited an increase in Mean Prestim events compared to control, **p = 0.01. After accounting for this increase in Mean Prestim events, *P[syt^{D-N}]* still displayed an increase in Mean (Async—Prestim)/Trace events compared to control, *p = 0.04.

(13.9 ± 3.1 pC, p = 0.82, Dunnett's correction) compared to control. Thus, blocking $Ca^{2+}$ binding by the $C_2A$ domain of synaptotagmin had no impact on asynchronous release, provided the electrostatic repulsion of the presynaptic membrane was maintained.

## Discussion

We investigated the role of $Ca^{2+}$ binding by the $C_2A$ domain of synaptotagmin 1 during neurotransmitter release at an intact synapse from *Drosophila* larvae by comparing two distinct mutant lines that both block $C_2A$ $Ca^{2+}$ binding. In the *P[syt^{D-N}]* mutant, binding is blocked by removing key negative charges required to coordinate $Ca^{2+}$ (Fig 1A [20, 32, 33]). This mutant displayed: no deficits in the amplitude of synchronous release (Fig 2A and 2B) or the number of fusion competent vesicles (Fig 3C), an increase in spontaneous release frequency (Figs 2E and 4D), and an increase in asynchronous release compared to control (Fig 4D–4F). These findings are consistent with previous reports [17–19, 42]. In addition, the paired pulse ratio was similar to control (Fig 3E), indicating no impact on release probability. In the *P[syt^{D-E}]* mutant, $Ca^{2+}$ binding is blocked by steric hindrance while maintaining the negative charge of the binding pocket (Fig 1A [4]). This mutant exhibited: a dramatic reduction in the amplitude of synchronous release (Fig 2A and 2B) and no change in spontaneous release frequency (Figs 2E and 4D), as seen previously [4]. As there was no change in the number of fusion competent vesicles (Fig 3C), the decrease in synchronous release could result from the decrease in release probability (Fig 3D and 3E). Importantly, we found no change in asynchronous release (Fig 4D–4F), in dramatic contrast to the *P[syt^{D-N}]* mutant.

Synaptotagmin serves as an electrostatic switch to trigger vesicle fusion events. Once $Ca^{2+}$ binds the negatively-charged $C_2A$ and $C_2B$ binding pockets, synaptotagmin's electrostatic repulsion of the negatively-charged presynaptic membrane (Fig 1A, top left) switches to electrostatic attraction (Fig 1A, top right and [4, 21, 22, 43]). This attraction then allows hydrophobic residues at the tips of the C2 domains to escape from the hydrophilic cytosol by penetrating into the hydrophobic core of the presynaptic membrane (Fig 1A, top right, grey residues and [44, 45]), favoring fusion *in vivo* [46, 47]. The two $C_2A$ mutations in this paper, which both block $Ca^{2+}$ binding, should be expected to have opposite effects on the amplitude of synchronous release, release probability, spontaneous release frequency, and asynchronous release, since they have an opposite impact on the charge of the $Ca^{2+}$-binding pocket.

When $Ca^{2+}$ binds to $C_2B$ during synchronous release, the *syt^{D-N}* mutation in $C_2A$ may participate in downstream effector interactions due to its decreased electrostatic repulsion of the presynaptic membrane (Fig 1A, bottom left, small arrows). Thus, the EJP amplitude in this mutant is as robust as in control both *in vivo* and in culture (Fig 2A and 2B and [17–20]). However, the mechanism may involve more than simply decreasing electrostatic repulsion, since mutating $C_2A$ aspartates to neutral alanines, which also removed the electrostatic repulsion, reduced IPSC amplitudes by 40% in cultured neurons [48]. This disparity may reflect differences between excitatory (the aspartate to asparagine studies) and inhibitory (the aspartate to alanine study) synapses. Or perhaps the more similar molecular structures of aspartate and asparagine, vs. alanine, supports full function when aspartate residues are replaced. Compared to the *syt^{D-N}* mutation, the *syt^{D-E}* mutation would have the opposite impact in terms of electrostatic switch function. Since electrostatic repulsion remains intact (Fig 1A, bottom right, large arrows), the *syt^{D-E}* mutation cannot participate in downstream membrane interactions [4]. Therefore, the release probability is significantly reduced (Fig 3D and 3E), fewer vesicles are triggered to fuse, and the EJP amplitude is dramatically reduced (Fig 2A and 2B and [4]). For spontaneous release, the *syt^{D-N}* and *syt^{D-E}* mutations should also result in opposite effects. At rest, SNARE proteins mediate constitutive vesicle-target membrane fusion reactions

throughout cells [49]. At the synapse, synaptotagmin 1 is required to help prevent aberrant SNARE-mediated spontaneous fusion events [6, 35, 36, 38, 50]; *syt*$^{D-N}$ mutations in either C2 domain result in an increase in the rate of spontaneous release [3, 17]. Importantly, the rate of spontaneous release is not increased by the *syt*$^{D-E}$ mutation in C$_2$A (Fig 2E and [4]). Thus, the decreased electrostatic repulsion between the C$_2$A domain of synaptotagmin and the presynaptic membrane in the *syt*$^{D-N}$ mutation increases spontaneous fusion events, while the maintenance of electrostatic repulsion in the *syt*$^{D-E}$ mutation prevents them.

Synaptotagmin null mutants show an increase in asynchronous release, indicating that the presence of synaptotagmin 1 inhibits aberrant asynchronous fusion events [15, 16]. Since this ability was maintained in a mutant lacking the C$_2$B domain, the C$_2$A domain was postulated to provide the inhibition of asynchronous fusion [16]. In addition, various *syt*$^{D-N}$ mutations that blocked Ca$^{2+}$ binding by the C$_2$A domain in either synaptotagmin 1 or Doc2 (another C2 domain based Ca$^{2+}$ sensor) resulted in increased asynchronous release [17, 51]. Together, these results led to the inhibition hypothesis: that Ca$^{2+}$ binding by the C$_2$A domain of synaptotagmin (or Doc2) directly inhibited asynchronous fusion events. However, the differential effects on asynchronous release between the two Ca$^{2+}$-binding mutants presented in this study are not consistent with this hypothesis. Rather, our findings are consistent with synaptotagmin's role as an electrostatic switch. In the *syt*$^{D-N}$ mutation, the removal of electrostatic repulsion by C$_2$A mimics constitutively bound Ca$^{2+}$ [18]. This would trigger fusion events for a longer period of time following Ca$^{2+}$ influx, resulting in increased asynchronous release. Since blocking Ca$^{2+}$ binding with our *syt*$^{D-E}$ mutation (which maintains electrostatic repulsion) did not result in a similar increase, our data demonstrate that the increase in asynchronous release caused by the *syt*$^{D-N}$ mutation was an unintended consequence of removing the electrostatic repulsion.

While our findings refute the inhibition hypothesis, all of the disparate changes in synchronous, spontaneous, and asynchronous release seen in both synaptotagmin null and C$_2$A domain mutants can be explained by a spatial competition hypothesis: synaptotagmin and an asynchronous Ca$^{2+}$ sensor spatially compete to regulate SNARE-mediated fusion. In null mutants, synaptotagmin 1 is no longer present to trigger synchronous release and clamp spontaneous release. Thus, SNARE-mediated spontaneous release increases, and the asynchronous Ca$^{2+}$ sensor has unimpeded access to SNARE complexes to trigger increased asynchronous release. The competition hypothesis is also consistent with results from the different Ca$^{2+}$ binding mutations. In the C$_2$A *syt*$^{D-N}$ mutation, the presence of synaptotagmin 1 blocks access to the SNARE complexes, but its function has been altered. This mutant C$_2$A domain, in combination with an intact C$_2$B domain, would support nearly normal synchronous release since the mutation results in an ostensibly constitutively Ca$^{2+}$-bound state. In addition, this mutation would be expected to trigger both the increase in spontaneous release and the increase in asynchronous release we observe. Indeed, the Ca$^{2+}$ dose-response curve for the C$_2$A *syt*$^{D-N}$ mutation was shifted to the left compared to control; the *syt*$^{D-N}$ mutation triggered more robust release at lower [Ca$^{2+}$] than control both *in vitro* [18] and *in vivo* [17]. This apparent increased Ca$^{2+}$ affinity would cause the *syt*$^{D-N}$ mutation to mimic an asynchronous sensor, as a higher Ca$^{2+}$ affinity is characteristic of asynchronous sensors [7]

Here we provide novel insights into synaptotagmin function during asynchronous neurotransmitter release. We report opposing results in two C$_2$A mutations of synaptotagmin that block Ca$^{2+}$ binding using distinct mechanisms. Our comparison of these two mutations demonstrates that by mimicking Ca$^{2+}$ binding, the *syt*$^{D-N}$ mutation can now act as an asynchronous sensor. Importantly, we disprove the current inhibition hypothesis regarding asynchronous neurotransmitter release. Instead, the spatial competition hypothesis can

account for these opposing effects and provides insight into an alternative mechanism regarding asynchronous neurotransmission.

## Supporting information

**S1 Raw Image.**
(PDF)

## Acknowledgments

The authors would like to acknowledge Dr. Motojiro Yoshihara for providing the *Drosophila* line containing the *P[UAS-syt 1$^{C2A-D3,4N}$]* mutation, Dr. Richard Daniels and Dr. Stefan Pulver for their assistance with *Drosophila* two-electrode voltage clamp physiology, and the CSU statistics department for their assistance, particularly Dr. Ann M. Hess.

## Author Contributions

**Conceptualization:** Mallory C. Shields, Noreen E. Reist.

**Data curation:** Mallory C. Shields, Matthew R. Bowers.

**Formal analysis:** Mallory C. Shields.

**Funding acquisition:** Mallory C. Shields, Noreen E. Reist.

**Investigation:** Mallory C. Shields, Matthew R. Bowers, Hannah L. Kramer, McKenzie M. Fulcer, Lara C. Perinet, Marissa J. Metz.

**Methodology:** Mallory C. Shields, Noreen E. Reist.

**Project administration:** Noreen E. Reist.

**Resources:** Mallory C. Shields, Noreen E. Reist.

**Supervision:** Mallory C. Shields, Noreen E. Reist.

**Validation:** Mallory C. Shields.

**Visualization:** Mallory C. Shields.

**Writing – original draft:** Mallory C. Shields, Noreen E. Reist.

**Writing – review & editing:** Mallory C. Shields, Matthew R. Bowers, Noreen E. Reist.

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
