## [Decision Letter · Decision Letter 0]

23 Mar 2020

PONE-D-20-04881

The role of the C2A domain of synaptotagmin 1 in asynchronous neurotransmitter release

PLOS ONE

Dear Dr. Reist,

Thank you for submitting your manuscript to PLOS ONE. After careful consideration, we feel that it has merit but does not fully meet PLOS ONE’s publication criteria as it currently stands. Therefore, we invite you to submit a revised version of the manuscript that addresses the points raised during the review process.

Please be sure to respond to the following concerns of reviewer 1:

1. For each experiment specify the genotypes in the methods and in each figure legend. In particular, what Gal4 driver was used to drive the syt transgenes? 

 2. Specify the extracellular calcium concentration used in the electrophysiology experiments in each figure legend.

3. Explain why TEVC was used in Fig. 4 rather than current clamp. It’s not clear why TEVC is needed to assess miniature events.

4. Consider shortening the discussion.

5. In addition please consider carefully all the comments of both reviewers and provide a response to each one with your revised submission.

We would appreciate receiving your revised manuscript by May 07 2020 11:59PM. To enhance the reproducibility of your results, we recommend that if applicable you deposit your laboratory protocols in protocols.io, where a protocol can be assigned its own identifier (DOI) such that it can be cited independently in the future. For instructions see: http://journals.plos.org/plosone/s/submission-guidelines#loc-laboratory-protocols

We look forward to receiving your revised manuscript.

Kind regards,

William D Phillips

Academic Editor

PLOS ONE

Journal Requirements:

Reviewers' comments:

Reviewer's Responses to Questions

**Comments to the Author**

1. Is the manuscript technically sound, and do the data support the conclusions?

Reviewer #1: Yes

Reviewer #2: Yes

2. Has the statistical analysis been performed appropriately and rigorously? 

Reviewer #1: Yes

Reviewer #2: Yes

3. Have the authors made all data underlying the findings in their manuscript fully available?

Reviewer #1: Yes

Reviewer #2: Yes

4. Is the manuscript presented in an intelligible fashion and written in standard English?

Reviewer #1: Yes

Reviewer #2: Yes

5. Review Comments to the Author

Reviewer #1: Shields et al. undertake an important study to re-visit the role of calcium binding to the C2A domain of the calcium sensor synaptotagmin. Previous research had suggested that while calcium binding to the C2B domain was needed for evoked release, binding to the C2A was dispensable but was needed to suppress asynchronous release. However, all of this data was based on C2A D-N mutations, which may mimic calcium binding. The authors test this idea using C2A D-E mutations, which conserved the neutral charge. In a short series of electrophysiological experiments, the authors demonstrate that synaptic transmission behaves very differently depending on whether the C2A D-N vs D-E mutations are used, and provide convincing evidence that the previous results using the D-N mutations were likely an artifact of losing the negative charge in C2A. Together, the author propose that both C2A and C2B and needed for proper evoked transmission, and that C2A does not have a role in suppressing asynchronous release.

Overall this is an important and rigorous study and the results provide evidence that will help to clear up misconceptions in the field and establish a foundation for future studies. I do have several suggestions, however, to improve the clarity and strengthen the conclusions.

Issues to consider:

1. Ideally all synaptotagmin transgenes would be inserted into the same attP targeting site to ensure consistent expression throughout development. This concern is mitigated in this study by the western blot data shown in Fig. 1B with similar expression levels observed. Nonetheless, it would be a great resource for future studies to have a clean system of all syt transgenes cloned into the same UAS backbone and inserted into the same attP site. The authors should also show baseline physiology in their yw control genotype to ensure there are no major differences compared to their syt WT expression control.

2. It was not clear anywhere in the manuscript what the genotypes were for the various experiments. In particular, what Gal4 driver was used to drive the syt transgenes? This needs to be clearly stated in the methods and in each figure legend.

3. Electrophysiology: There are several improvements to the electrophysiological data that should be considered:

A. No areas in the methods or figure legends could I find the extracellular calcium concentration used in the experiments. Obviously this is a key parameter for assessing synaptic transmission. This should be clearly stated in each figure.

B. The authors should consider recording in at least 2-3 different calcium conditions to assess evoked transmission. A low and high condition (0.4 mM, 1.8 mM), ideally in TEVC.

C. The authors use current clamp approaches in Figures 2-3, then switch to TEVC for no apparent reason. At minimum the authors should explain why TEVC was used in Fig. 4, since all they are measuring are mEPSC events. TEVC is superior over current clamp for evoked measurements (Fig. 2) because of nonlinear summation, but it’s not clear why TEVC is needed to assess miniature events.

D. Paired pulse: The data presented in the study is indeed consistent with reduced Pr in syt D-E. However, one question is whether NMJs espressing syt D-E is functioning like a control NMJ at a reduced Pr, or if there are any mechanistically distinct features. The authors should consider recording from their wild type syt control (and yw) at reduced extracellular calcium to adjust release to the same baseline state (~8 mV), then perform PPF to determine if syt D-E is simply behaving like a WT NMJ operating at reduced Pr.

E. The hypertonic sucrose experiments show that releaseable pools are similar between the 3 genotypes compared. These sucrose experiments can be quite variable however, depending on where the sucrose is applied, etc. An improved method the authors should consider is to measure the readily releasable vesicle pool by stimulating at high frequency at elevated calcium in TEVC. This would provide an independent measure of similar pools as well.

4. The discussion is quite long. Much of this can be repurposed to a review, which would be timely to discuss the updated insights of this work and others for the field. However, for this manuscript, it should be condensed and limited to the major points that derive from this specific study, perhaps 5-6 paragraphs.

Reviewer #2: The manuscript by Shields et al. presents a panoply of synaptic transmission experiments to demonstrate differences between the sytD-E and sytD-N C2A mutants. Of major interest, the former mutant does not increase asynchronous release, thus supporting the electrostatic switch model of synaptotagmin function and disproving some prior models. To address the known increase in spontaneous minis induced by syt knockout, the authors suggest that other Ca2+ sensors are recruited. The experimental results are clearly presented and the analysis is consistent with the new results. Therefore, this is a valuable contribution to understanding how synaptotagmin functions as a Ca2+ sensor, but perhaps not as clamp as previously envisioned.

Minor concerns:

1. An experimental artifact usually refers to a result being incorrect due to some unforeseen factor. However, here it refers to a reinterpretation of valid results due to new experimental results. An analogous example would be the Hodgkin Huxley treatment of sodium channels; nobody refers to their model as being an artifact now that we know more from single channel recording. Rather, the functioning of sodium channels is now reformulated to take into account that inactivation is faster than activation. Therefore, the authors should dispel with the use of the word “artifact” in the context used here and refer to the reformulation or reassessment (or some other similar concept) of synaptotagmin function.

2. Aren’t the facts that the sytD-E mutant is less effective and the sytD-N is partially active due to its mimicking of Ca2+ binding sufficient to explain the data on their own? If so, the emphasis on the spatial competition hypothesis in the last sentence is puzzling. If I am missing something, perhaps this aspect of the presentation could be clarified.

3. Can the authors comment on the effects of sytD--E mutation of the C2B domain?

6. PLOS authors have the option to publish the peer review history of their article (what does this mean?). If published, this will include your full peer review and any attached files.

Reviewer #1: Yes: Dion Dickman

Reviewer #2: No

---

## [Author Response · Author response to Decision Letter 0]

20 Apr 2020

We would like to thank the reviewers for their thoughtful and constructive critiques of our manuscript. We have responded to each comment below and believe the edited the manuscript is now a clearer presentation of our findings. 

Reviewer #1:

“1. Ideally all synaptotagmin transgenes would be inserted into the same attP targeting site to ensure consistent expression throughout development. This concern is mitigated in this study by the western blot data shown in Fig. 1B with similar expression levels observed. Nonetheless, it would be a great resource for future studies to have a clean system of all syt transgenes cloned into the same UAS backbone and inserted into the same attP site.”

Although identical insertion sites are desirable for new experimental lines, this study was designed to reconcile data from two previously published mutants, P[sytD-N], Yoshihara et al 2010 and P[sytD-E], Streigel et al. 2012. Therefore, we used the specific lines generated during the previous studies. 

“The authors should also show baseline physiology in their yw control genotype to ensure there are no major differences compared to their syt WT expression control.”

Our transgenic sytWT control line provides approximately a 70% rescue of evoked transmitter release (Fig. 4, Mackler and Reist, 2001, J Comp Neurol 436:4-16). Multiple independent lines were tested at that time and all provided approximately the same level of evoked transmitter release (unpublished observation). As this has been previously established, we did not repeat the comparison here. We have edited the methods section to clarify this point. 

“2. It was not clear anywhere in the manuscript what the genotypes were for the various experiments. In particular, what Gal4 driver was used to drive the syt transgenes? This needs to be clearly stated in the methods and in each figure legend.”

Thank you for pointing out this oversight. We have edited the methods section to specify which Gal4 driver was used to express our UAS-transgenes and clarify the specific genotypes used. As the same genotypes were used in every figure, we use the abbreviated genotypes (now specifically defined in the methods section) in the figure legends. 

“3. Electrophysiology: There are several improvements to the electrophysiological data that should be considered:

A. No areas in the methods or figure legends could I find the extracellular calcium concentration used in the experiments. Obviously this is a key parameter for assessing synaptic transmission. This should be clearly stated in each figure.”

Thank you for this suggestion, as we agree that the Ca2+ concentration used should be easily assessable. All recordings were made in HL3.1 containing 1.0 mM Ca2+. However, the Ca2+ concentration was only stated once when HL3.1 was first defined under the Immunoblotting section of the methods. To clarify this critical point, we now restate the Ca2+ level both in the Electrophysiological experiments and analyses section of the methods and in each figure legend. 

“B. The authors should consider recording in at least 2-3 different calcium conditions to assess evoked transmission. A low and high condition (0.4 mM, 1.8 mM), ideally in TEVC.”

Analysis of evoked release across a range of Ca2+ concentrations has already been published for both the P[sytD-N] fly line (Yoshihara et al 2010) and the P[sytD-E] fly line (Streigel et al. 2012). 

“C. The authors use current clamp approaches in Figures 2-3, then switch to TEVC for no apparent reason. At minimum the authors should explain why TEVC was used in Fig. 4, since all they are measuring are mEPSC events. TEVC is superior over current clamp for evoked measurements (Fig. 2) because of nonlinear summation, but it’s not clear why TEVC is needed to assess miniature events.”

As current clamp is fast and efficient, we used it for all experiments where it could measure the desired parameter: synchronous evoked release, mEPSP amplitude, mEPSP frequency, paired pulse and hypertonic sucrose stimulation, presented in figures 1-3. In addition, it was important to measure synchronous release by current clamp to determine the efficacy of our TEVC recordings and limit inclusion to those that attained a minimum of 90% clamp. Note, the same patterns were observed for the synchronous evoked release in both current clamp (Fig 2) and TEVC (Fig 4F). To investigate asynchronous release, TEVC is required. Individual mEPSC events that occur within 20-300 msec after the synchronous evoked response (the asynchronous response) cannot be resolved by current clamp in any line with a robust synchronous response due to non-linear summation. We have clarified in the text why the switch to TEVC was necessary. 

“D. Paired pulse: The data presented in the study is indeed consistent with reduced Pr in syt D-E. However, one question is whether NMJs espressing syt D-E is functioning like a control NMJ at a reduced Pr, or if there are any mechanistically distinct features. The authors should consider recording from their wild type syt control (and yw) at reduced extracellular calcium to adjust release to the same baseline state (~8 mV), then perform PPF to determine if syt D-E is simply behaving like a WT NMJ operating at reduced Pr.”

Since Pr can explain the changes that we see and synaptotagmin is known to impact Pr, we believe our current experiments are appropriate for this publication. We would like to thank the reviewer for this interesting suggestion to probe alternative mechanisms. We may pursue such inquiry once the restrictions on our research due to Covid-19 are lifted.

“E. The hypertonic sucrose experiments show that releasable pools are similar between the 3 genotypes compared. These sucrose experiments can be quite variable however, depending on where the sucrose is applied, etc. An improved method the authors should consider is to measure the readily releasable vesicle pool by stimulating at high frequency at elevated calcium in TEVC. This would provide an independent measure of similar pools as well.”

Since high frequency stimulation relies on Ca2+ dependent vesicle fusion, which is severely affected in our D-E mutants, this approach has so far proven ineffective in our previous attempts to measure the readily releasable pool of vesicles. As such, we chose to focus on the pool of vesicles capable of non-Ca2+ dependent fusion.

“4. The discussion is quite long. Much of this can be repurposed to a review, which would be timely to discuss the updated insights of this work and others for the field. However, for this manuscript, it should be condensed and limited to the major points that derive from this specific study, perhaps 5-6 paragraphs.”

We have shortened the discussion by >20% and limited it to essential points. It is now 6 paragraphs.

Reviewer #2: 

“Minor concerns:

1. An experimental artifact usually refers to a result being incorrect due to some unforeseen factor. However, here it refers to a reinterpretation of valid results due to new experimental results. An analogous example would be the Hodgkin Huxley treatment of sodium channels; nobody refers to their model as being an artifact now that we know more from single channel recording. Rather, the functioning of sodium channels is now reformulated to take into account that inactivation is faster than activation. Therefore, the authors should dispel with the use of the word “artifact” in the context used here and refer to the reformulation or reassessment (or some other similar concept) of synaptotagmin function.”

That the D-to-N substitutions mimic Ca2+ binding while concurrently blocking that binding was, at the time, an unforeseen factor. However, we see your point. We have rephrased our manuscript to state that our new data requires a new interpretation the previous studies’ findings.

“2. Aren’t the facts that the sytD-E mutant is less effective and the sytD-N is partially active due to its mimicking of Ca2+ binding sufficient to explain the data on their own? If so, the emphasis on the spatial competition hypothesis in the last sentence is puzzling. If I am missing something, perhaps this aspect of the presentation could be clarified.”

While functional results in the syt-C2A-D-N mutant can be explained by this mutation mimicking Ca2+ binding, the hypothesis that Ca2+ binding by C2A regulates an asynchronous sensor was based on results from both C2A Ca2+ binding mutants AND sytnull mutants. Mimicking Ca2+ binding by the sytC2A-D-N mutant cannot account for an increase in asynchronous release observed in sytnull mutants. Therefore, we propose the spatial competition hypothesis which can account for the increase in asynchronous release in all synaptotagmin 1 mutants reported to date.

“3. Can the authors comment on the effects of sytD-E mutation of the C2B domain?”

These experiments are beyond the scope of this study. However, we will be pursuing this direction.

---

## [Editor Report · Decision Letter 1]

27 Apr 2020

The role of the C2A domain of synaptotagmin 1 in asynchronous neurotransmitter release

PONE-D-20-04881R1

Dear Dr. Reist,

We are pleased to inform you that your manuscript has been judged scientifically suitable for publication and will be formally accepted for publication once it complies with all outstanding technical requirements.

With kind regards,

William D Phillips

Academic Editor

PLOS ONE
---

## [Editor Report · Acceptance letter]

1 May 2020

PONE-D-20-04881R1 

The role of the C_2_A domain of synaptotagmin 1 in asynchronous neurotransmitter release 

Dear Dr. Reist:

I am pleased to inform you that your manuscript has been deemed suitable for publication in PLOS ONE. Congratulations! Your manuscript is now with our production department. 

With kind regards,

on behalf of

Dr. William D Phillips 

Academic Editor

PLOS ONE